# Hydrogen Sulfide in the Oxidative Stress Response of Plants: Crosstalk with Reactive Oxygen Species

**DOI:** 10.3390/ijms25031935

**Published:** 2024-02-05

**Authors:** Zhiya Liu, Yayu Liu, Weibiao Liao

**Affiliations:** College of Horticulture, Gansu Agricultural University, 1 Yinmen Village, Anning District, Lanzhou 730070, China; lzyaaa1127@163.com (Z.L.); liuyayu199809@163.com (Y.L.)

**Keywords:** redox signaling molecule, persulfidation, abscisic acid, oxidative stress, co-operative interaction

## Abstract

Growing evidence suggests that exposure of plants to unfavorable environments leads to the accumulation of hydrogen sulfide (H_2_S) and reactive oxygen species (ROS). H_2_S interacts with the ROS-mediated oxidative stress response network at multiple levels. Therefore, it is essential to elucidate the mechanisms by which H_2_S and ROS interact. The molecular mechanism of action by H_2_S relies on the post-translational modification of the cysteine sulfur group (-SH), known as persulfidation. H_2_S cannot react directly with -SH, but it can react with oxidized cysteine residues, and this oxidation process is induced by H_2_O_2_. Evidently, ROS is involved in the signaling pathway of H_2_S and plays a significant role. In this review, we summarize the role of H_2_S-mediated post-translational modification mechanisms in oxidative stress responses. Moreover, the mechanism of interaction between H_2_S and ROS in the regulation of redox reactions is focused upon, and the positive cooperative role of H_2_S and ROS is elucidated. Subsequently, based on the existing evidence and clues, we propose some potential problems and new clues to be explored, which are crucial for the development of the crosstalk mechanism of H_2_S and ROS in plants.

## 1. Introduction

Hydrogen sulfide (H_2_S) is a toxic and harmful gas. The synthesis of endogenous H_2_S was first reported in tobacco by Harrington and Smith [1]. Since then, many studies have studied the function and properties of H_2_S in plants. When following the fate of ^35^S-labeled L-cysteine, L-cysteine desulfhydrase (L-CDes) was found to catalyze the degradation of L-cysteine, producing H_2_S, pyruvate, and ammonium [1]. With the discovery of L-CDes, the activity of L-CDes was detected in an increasing number of species, and it became known as one of the most important sources of H_2_S enzymes in plants [2,3]. Then, the genes encoding these H_2_S-producing enzymes became the main targets of research. Four key genes encoding cysteine desulfurases have been reported in *Arabidopsis thaliana* [4,5]. Similarly, these genes have been located and characterized in other plants [2,3,6]. H_2_S is the third gas signaling molecule after NO and CO, according to extensive study; it is essential to plant growth and development and to a plant’s ability to respond to a variety of biotic and abiotic challenges [7,8,9,10].

Over the past decade, there has been a greater interest in the “new role” of H_2_S in plants. Numerous studies have shown that H_2_S acts as a gaseous signaling molecule that is ubiquitous in enhancing plant adaptation/tolerance to various abiotic stresses, including drought, salt, chilling, osmotic stress, and others [10,11,12,13]. Obviously, these abiotic stresses are always followed by reactive oxygen species (ROS) production, and the mode of action of H_2_S in the stress response is usually related to oxidative stress. Previous studies demonstrated that H_2_S exhibits antagonistic effects with ROS in oxidative stress [8,14,15]. It was generally assumed that H_2_S is a natural regulator of plants in response to oxidative stress and protects plants from oxidative damage caused by ROS accumulation. However, there is growing evidence for both antagonistic and synergistic roles of H_2_S and ROS in regulating plant responses to challenging conditions [9,14,16], as the involvement of H_2_O_2_ was found in studies of H_2_S-mediated post-translational modifications (PTMs) [17]. H_2_S is unable to react directly with protein cysteine residues (-SH) to form persulfides; instead, it reacts only with oxidized sulfinic acid (-SOH), and this oxidation process must occur in the presence of H_2_O_2_ (Figure 1). As such, an appropriate level of H_2_O_2_ is required to facilitate H_2_S signaling cascades, especially under oxidative stress [18]. Therefore, ROS are involved in H_2_S-mediated oxidative post-translational modifications as secondary signals, either antagonistically or synergistically. Currently, several mechanisms for the interaction of H_2_S and ROS in oxidative stress have been proposed [19,20,21,22]. However, the relationship between H_2_S and ROS remains to be fully elucidated.

Additionally, plant stress responses are mediated by spatiotemporal coordination between ROS and other signals, such as ABA. Numerous studies have shown interconnections between H_2_S and/or ROS and ABA in plants exposed to unfavorable environments, such as response to drought stress and the regulation of stomatal movement [17,23,24]. Proteomic data show that the ABA signaling network involves the persulfidation of multiple proteins and that persulfidation is triggered in response to ABA [25]. Meanwhile, both H_2_S and H_2_O_2_ are involved in ABA-induced stomatal closure, suggesting a complex cross-reaction between them.

In the past decades, H_2_S has been mostly known to prevent excessive ROS accumulation under stress, but recent studies have revealed that it can also regulate cellular homeostasis by actively cooperating with ROS under oxidative stress. Although Chen et al. and Zhang et al. reviewed the multifunctionality of H_2_S in plants in response to stress [7,14], to date, there is very limited systematic elaboration on the positive crosstalk between H_2_S and ROS signaling molecules under oxidative stress. Therefore, this review provides a systematic review of the crosstalk mechanisms of H_2_S and ROS in plant cells under oxidative stress. Emphasis is placed on the active role of persulfidation in oxidative stress and the mechanisms of crosstalk between H_2_S and ROS at multiple levels. Thus, this review will increase our understanding of the interactions between H_2_S and ROS, especially the active roles of H_2_S and ROS in response to oxidative stress.

## 2. Biosynthesis of Endogenous H_2_S in Plants

Initially, H_2_S was reported to be a toxic gas. Following a lengthy process of research, it was found that H_2_S can be synthesized in plants [26], and the sulfur (S) component is primarily derived from sulfate ions taken up by plant roots. Adenosine triphosphate (ATP) activates the sulfate transporter, which absorbs extracellular sulfate. Adenosine 5′-phosphate sulfate (APS) is produced under the action of ATP-sulfurylase. APS REDUCTASE (APR) subsequently reduces APS to sulfite, which is subsequently reduced to H_2_S by sulfite reductase (SiR) [27]. This process relies on the photosynthetic pathway of chloroplasts. Later, it was shown that H_2_S release could be detected when exogenous sources provided bisulfite, sulfate, sulfite, or L-cysteine [28]. This suggests that H_2_S can not only be endogenously synthesized in plants, but that it can also be released in excess to ensure H_2_S homeostasis in the plant, thereby promoting plant growth and development and avoiding toxic damage.

It was subsequently demonstrated that tobacco’s release of H_2_S was caused by cysteine degradation. L-cysteine desulfhydrase (also known as L-CDes; EC 4.4.1.1) catalyzes the breakdown of L-cysteine to produce H_2_S, pyruvate, and ammonium [1]. L-CDes is one of the most important sources of H_2_S enzyme in plants and has been discovered to be widely distributed in a variety of plants. Subsequently, Riemenschneider et al. characterized and isolated an *Escherichia coli* (*E. coli*) protein (YedO) that has DCD activity [5]. Based on this result, D-cysteine desulfhydrase (D-CDes) was identified in *A. thaliana* with a high degree of homology to YedO [29]. This is considered to be another important enzyme for the production of H_2_S. In plant cells, L-CDes and D-CDes are located in the cytoplasm and mitochondria. Cysteine degradation in different subcellular compartments requires the participation of the coenzyme 5′-pyridoxal phosphate (PLP), which produces H_2_S, ammonia, and pyruvate in a 1:1:1 ratio (Figure 2). In addition, chloroplast Nifs-like proteins have been identified in *A. thaliana* and are PLP-dependent enzymes [30]. They have cysteine desulfurase activity and catalyze the conversion of cysteine to alanine and elemental S [31]. This acts as another pathway for the production of H_2_S. It appears that cysteine (Cys) plays a key role in the metabolism of H_2_S. Cysteine metabolism and H_2_S metabolism are also correlated, either directly or indirectly [13]. Moreover, another enzyme, O-acetylserine (thiol) lyase (OASTL), which plays a major role in the final step of cysteine synthesis, catalyzes the binding of o-acetylserine (OAS) to H_2_S to form Cys. OASTL family genes have been identified in *Arabidopsis* and are localized in the cytosol, plastids, and mitochondria [32,33]. Among them, the CYSTEINE SYNTHASE-LIKE protein [34], encoded by the At5g28030 gene, is involved in the degradation rather than the biosynthesis of L-Cys. Thus, CS-LIKE was considered a novel L-CDes and given the designation DES1 [32]; its functions have been extensively explored. It has been tacitly recognized that cytoplasmic DES1 is the major L-Cys in *Arabidopsis* [7]. However, Kurmanbayeva et al. supported the notion that OASTLA, OASTLB, and OASTLC, located in the cytoplasm, chloroplasts, and mitochondria, are the main components of desulfurase activity in *Arabidopsis thaliana*, degrading not only L-Cys but also L-SeCys [35]. Therefore, the enzymatic reaction of OASTL is a reversible process that produces endogenous H_2_S, confirming the speculation that OAS-TL has bidirectional catalytic activity; this also coincides with Liu et al.’s review [33]. Previous research identified three genes encoding OASTL (*AtcysCl, AtcysDl*, and *AtcysD2*) from *A. thaliana* [36]. Complementation assays with *E. coli* confirmed that *AtcysCl* is actually a β-cyanoalanine synthetase (β-CAS) and not a cysteine synthetase. In mitochondria, β-CAS catalyzes the reaction of L-Cys with HCN to synthesize β-cyanoalanine and H_2_S [27,36].

One strategy that has been proposed is the use of carbonyl sulfide (COS) as a donor of H_2_S. COS is a gaseous component of the atmosphere that can be taken up by plants [37]. Small molecules of COS pooled in plants are rapidly hydrolyzed in the presence of carbonic anhydrase (CA) to produce H_2_S (Figure 2). Chauhan et al. reported that enzyme-activated COS donors are able to induce protein persulfuration and alleviate oxidative stress [38]. Therefore, these results suggest that CA is a new enzyme that produces H_2_S in plants, which is a new molecular mechanism for the H_2_S production pathway. Moreover, amino acid-derived N-thiocarboxylic anhydrides [39] have recently been found in animals to be capable of releasing H_2_S in the presence of CA [39]. However, this has not yet been reported in plants; it is likely to be a new mechanism for producing endogenous H_2_S and will be a new direction for future research (Figure 2).

## 3. Post-Translational Modification (PTM) of Cysteine Residues by H_2_S under Oxidative Stress

The signaling perception and transduction mechanisms of H_2_S are essential to its function. Although there are several possible mechanisms, current findings suggest that the main mode of H_2_S signaling is the persulfidation modification of protein cysteine residues (RSH) by H_2_S [9,14]. Protein persulfidation, also known as S-sulfhydration, is a reversible process in which RSH is oxidized to a persulfide thiol (RSSH) group. Several reports have indicated that the redox-based reversible post-translational modification not only regulates plant growth and developmental processes but also serves as a protective mechanism for H_2_S under oxidative stress [40,41,42]. In the following sections, we highlight the molecular mechanisms of persulfidation in the regulation of oxidative stress.

There has been significant progress in recent years in the study of the multiple biological functions of H_2_S, which is recognized as a natural regulator of oxidative stress responses in plants [43,44]. Such stress causes an increase in the production of ROS (H_2_O_2_), which is sensed and transmitted as a secondary signal in plant cells, and these molecules can lead to redox-dependent post-translational modifications of proteins. It is apparent that H_2_S and its mediated persulfidation play a central role in this process, through its unique chemical properties [45]. Under oxidative stress conditions, the generated ROS (H_2_O_2_) can lead to cysteine sulfur (Cys) oxidation, which can alter the structure of the target proteins and, in severe cases, even degrade them (Figure 1B). However, H_2_S prevents the oxidation of Cys residues in target proteins through persulfidation [46]. Thus, persulfidation has been suggested as a protective mechanism against oxidative stress. For example, under salt stress, H_2_S triggers PMA1 to sustain the K^+^/Na^+^ balance by means of persulfidation [11]; nitrate reductase 2 (NIA2) persulfidation is involved in OsLCD1-regulated drought tolerance in rice [22]. Proteomic analysis showed that the enriched motif containing the most persulfidated proteins (11) is involved in the cellular response to oxidative stress in *Arabidopsis* under drought stress (Table 1), containing numerous antioxidant enzymes [47]. A similar situation occurs in tomato, where, in response to oxidative stress induced by the phytotoxicity of CuO NPs, H_2_S responds to stress by nondirectionally regulating the activities of POD, CAT, and APX by means of persulfidation [48]; these effects seem to be the consequence of increased ROS contents, especially that of H_2_O_2_. The first step in persulfidation does require a signal from H_2_O_2_ via S-sulfenylation [46]. Many proteins have been identified as S-sulfenylation targets under H_2_O_2_ stress, and the conserved oxidation-sensitive cysteine (AtMAPK4 Cys181) in *Arabidopsis* mitogen-activated protein kinase (MAPK) plays an important role in the kinase mechanism [49], demonstrating that it is probably involved in the next step of the redox reaction (persulfidation). However, in the research on the mechanism of MPK4 persulfidation regulating cold stress, eight Cys sites were identified (including Cys181). Curiously, the persulfidation signals did not disappear after mutation of the sites [50], which may be due to the persulfidation of MPK4 occurring on two or more Cys residues; this specific mechanism needs to be further explored. H_2_S enhances the level of S-sulfhydration of His-Csa5G156220 and His-Csa5G157230 (bHLH transcription factors) and the binding activity to the promoter of Csa6G088690, a key synthase for CuC production, thereby enhancing cucumber stress tolerance [12]. Moreover, a recent study revealed that *Arabidopsis* G6PD6 Cys159 and tomato G6PDC Cys155 undergo persulfidation, resulting in a smaller spatial distance between lysine (K)491 and K475 in the G6PD tetramer, enhanced G6PD oligomerization, and the promotion of cytoplasmic G6PD activity, which, in turn, prevent oxidative damage [51] (Table 1).

Autophagy is a macro-molecular degradation pathway involving endoplasmic reticulum (ER) stress and is activated in response to stress [52]. ATG18a is a core autophagic component necessary for reticulophagy during cellular autophagy and endoplasmic reticulum (ER) stress. Aroca et al. revealed that reversible persulfidation modification of Cys103 activates the ability of ATG18a to bind specific phospholipids and inhibits the maturation of autophagosomes under endoplasmic reticulum stress, thereby ensuring that plants can carry out normal physiological responses [52]. Moreover, as a stress signaling molecule, ABA plays an indispensable role in the molecular mechanisms of H_2_S-mediated persulfidation in regulating oxidative stress. Laureano-Marín et al. showed that ABA-triggered persulfidation revealed autophagy-associated (ATG) protease-specific persulfidation, demonstrating that the negative regulation of autophagy by H_2_S is mediated through the persulfidation of AtATG4a Cys170 [53] (Table 1).

In the previous section, we summarized that photorespiration is a major pathway for the production of ROS, which can change the cellular redox balance. A large proteomic analysis showed that many of the proteins associated with photorespiration are also modified by cysteine redox PTMs (reviewed in [54], Table 2). Interestingly, persulfidation is the most broadly distributed PTM in the photorespiratory pathway [54,55]. A quantitative analysis of differentially persulfidated proteins under suppressed photorespiration (NPC) and adapted to air (APC) showed that 98.7% of target proteins were more persulfidated under NPC, and with higher levels of H_2_O_2_. When plants were transferred from NPC to APC, the level of persulfidation decreased dramatically, which may be associated with the increase in plant H_2_S content under NPC [55]. A gel analysis of the levels of persulfidation and sulfenylation revealed that *A. thaliana* showed a dramatic decrease in the S-sulfenylation response of the plant during the transition to photorespiration. However, after 3 d of domestication under normal air conditions, plant S-sulfenylation levels increased again, which was mainly regulated by a low H_2_O_2_ concentration [55]. Therefore, the protective effect of sulfide on plants under NPC is mediated by the regulation of H_2_O_2_ levels through reversible redox modifications of persulfidation and sulfenylation.

**Table 1 ijms-25-01935-t001:** Proteins of H_2_S mediated persulfidation in oxidative stress responses.

Proteins	Sites	Plant Species	Reference
Ascorbate peroxidase 1	Cys32	*Arabidopsis thaliana*	[56]
Cytosolic ascorbate peroxidase 1	Cys234	*Solanum lycopersicum*	[48]
Catalase 1	Cys168	*Solanum lycopersicum*	[48]
Percoxidase 5	Cys46, 61	*Solanum lycopersicum*	[48]
Plasma membrane H^+^-ATPase 1	Cys446	*Arabidopsis thaliana*	[11]
Open Stomata1 (Ost1)/Snf1-Related Protein Kinase 2.6	Cys131, 137	*Arabidopsis thaliana*	[57]
ABSCISIC ACID INSENSITIVE 4 (ABI4)	Cys250	*Arabidopsis thaliana*	[58]
L-cysteine desulfhydrase	Cys44, 205	*Arabidopsis thaliana*	[59]
Respiratory burst oxidase homolog protein	Cys825, 890	*Arabidopsis thaliana*	[59]
His-Csa5G156220	Unknown	*Cucumis sativus* L.	[12]
His-Csa5G157230	Unknown	*Cucumis sativus* L.	[12]
Nitrate Reductase 2	Unknown	*Oryza sativa* L.	[22]
Glucose-6-phosphate dehydrogenases 6	Cys159	*Arabidopsis thaliana*	[51]
Glucose-6-phosphate dehydrogenases C	Cys155	*Solanum lycopersicum*	[51]
Autophagy-related (ATG18a)	Cys103	*Arabidopsis thaliana*	[52]
Autophagy-related (ATG4)	Cys170	*Arabidopsis thaliana*	[53]
1-aminocyclopropane-1-carboxylic acid oxidases (ACOs)	Cys60	*Solanum lycopersicum*	[60]

## 4. Biosynthesis of Endogenous H_2_S in Plants’ Interaction between H_2_S and ROS in the Regulation of Oxidative Stress

As mentioned above, since H_2_S does not directly react with the sulfhydryl groups of proteins, it is generally accepted that protein persulfidation is the result of the reaction of H_2_S with the sulfhydryl group -SOH, which has been oxidized by H_2_O_2_ [14]. Therefore, the H_2_S signaling pathway requires the involvement of some level of H_2_O_2_. However, aside from the antagonistic roles discussed above, H_2_S and ROS can also act together in the stress response.

### 4.1. H_2_S Positively Regulates Oxidative Signal-Induced Stomatal Closure in Upstream and Downstream of H_2_O_2_

H_2_O_2_ is widely recognized as an important player in guard cell signaling, in which potential sources of H_2_O_2_ enzymes, including NADPH oxidases, amine oxidases, and peroxidases, play important roles [14,61,62]. The production of cytosolic ROS is primarily dependent on members of the NADPH oxidase respiratory burst oxidase homologues (RBOH), namely, RBOHD and RBOHF, which are engaged in the regulation of stomatal movement [62]. Scuffi et al. revealed that RBOHD and RBOHF are required during H_2_S-induced stomatal closure, and NADPHox acts downstream of H_2_S [63]. Thus, these results indicate that there is interplay between H_2_S and H_2_O_2_ in regulating stomatal closure.

Another messenger is phosphatidic acid (PA), which is produced by phospholipase D (PLD). Previous studies have shown that PLD binds to RBOHD and RBOHF and induces ABA-dependent ROS generation [64,65]. The members of the PLD multigene family, PLDα1 and PLDδ, are implicated in ABA-dependent stomatal closure, although only PLDδ is engaged in modulating ROS production [64,65]. However, in this example, it was revealed that H_2_S-induced H_2_O_2_ production requires PLDα1 but not PLDδ, contrary to the results for ABA. Interestingly, in *pldα1* and *pldδ*, H_2_S-dependent stomatal closure was interdicted, demonstrating that the process of H_2_S-mediated stomatal closure requires two PLD isoenzymes [63]. It was also discovered that H_2_S induces PA generation in the RBOHD-dependent pathway through the activity of PLDδ. The evidence shows that stomatal closure triggered by osmotic stress requires PLDδ and H_2_S, and that LCD acts downstream of PLDδ [66]. Apparently, osmotic stress leads to H_2_O_2_ accumulation in guard cells, which induces stomatal closure. However, the potential signaling network among the three signaling molecules PLDδ, H_2_S, and H_2_O_2_ and their interrelationships were clarified in another example. It was shown that NADPH oxidase-dependent H_2_O_2_ production is necessary in the induction of stomatal closure by osmotic stress, and H_2_O_2_ acts downstream of PLDδ and LCD [67]. The two signaling molecules, together with NADPH oxidase (RBOHD) and PLD (PLDδ), form a regulatory network to adjust upstream and downstream signals (Figure 3A).

Additionally, extracellular ATP (eATP) was thought to be involved in the regulation of stomatal closure. Wang et al. found that H_2_S influences stomatal closure by stimulating outward K^+^ currents in guard cells; the eATP and ABC transporters (*AtMRP4* and *AtMRP5*) are involved in this process, with H_2_O_2_ acting downstream of eATP as an essential component of stomatal closure [21]. Consequently, these results reveal the positive role of H_2_S and H_2_O_2_ in regulating stomatal movement, and H_2_S acts as an upstream regulator of H_2_O_2_ levels. Moreover, a recent study revealed that H_2_S and H_2_O_2_ are engaged in the signal transduction pathway for EBR-induced stomatal closure [68]. EBR promotes NADPH oxidase gene *AtrbohF*-dependent H_2_O_2_ production and further increases L-/D-CDes activity and H_2_S synthesis, which ultimately lead to stomatal closure. H_2_S is implicated in the EBR-induced stomatal closure process in *Arabidopsis* and acts as a downstream player of H_2_O_2_ in the process [68]. Interestingly, compared with the above results, the correlations between H_2_S and H_2_O_2_ are completely opposite. This could be the result of differences due to exogenous treatments and osmotic stress, or differences in the H_2_O_2_-producing pathways of RBOHD versus RBOHF, or others. Therefore, regardless of the mechanism, H_2_S and H_2_O_2_ play a synergistic role in regulating stomatal closure.

### 4.2. Positive Cooperation between H_2_S and H_2_O_2_ Signaling in the Stress Response

Previous studies generally concluded that H_2_O_2_ is a toxic byproduct of abiotic stress and inhibits plant growth and development, whereas H_2_S is considered a regulator to alleviate abiotic stress. However, H_2_S and H_2_O_2_ signaling also have a positive role in oxidative stress. The Put–H_2_S–H_2_O_2_ signaling pathway plays a pivotal role in promoting the uptake and accumulation of UV compounds in hull-less barley seedlings under UV-B stress. It is also involved in maintaining intracellular redox homeostasis [69]. Putrescine (Put) induces the synthesis of H_2_S through the activation of L-DES activity, stimulates the production of H_2_O_2_ dependent on the enzyme PM-NADPH oxidase, upregulates antioxidant enzyme activities, and increases the contents of AsA and GSH, thereby re-establishing redox homeostasis by reducing oxidative damage due to UV-B in barley plants [69]. Moreover, spermidine (Spd)-induced H_2_S acts downstream of H_2_O_2_ signaling in response to dehydration in white clover [70]. Spd induces H_2_S production by activating L/DCD activity, and the H_2_S signaling pathway is involved in Spd regulation of transcription factors and antioxidant systems to reduce oxidative damage in white clover [70]. The interaction between H_2_S and H_2_O_2_ also provides new insights into the complex signaling mechanisms underlying the salt tolerance response in *Arabidopsis*. It was indicated that H_2_S induces endogenous H_2_O_2_ production by modulating the activities of plasma membrane (PM) NADPH oxidase and glucose-6-phosphate dehydrogenase (G6PDH), which, in turn, enhance PM H^+^-ATPase activity, phosphorylation levels, and the expression of PM Na^+^/H^+^ antiporter protein levels, ultimately maintaining ionic homeostasis [19] (Figure 3B).

### 4.3. H_2_S Promotes ROS Bursts, Fine-Tuning Redox Homeostasis and ABA Signaling

In addition to the molecular mechanisms of the H_2_S and H_2_O_2_ interactions mentioned above, other signaling molecules also play significant roles in regulating redox homeostasis, such as ABA, ETH, IAA, SA, and Ca^2+^ signaling [57,60,71,72,73,74]. Among them, ABA, as an important phytohormone, has been extensively investigated in various oxidative stress responses. ABA plays a central role in stomatal closure and promoting plant adjustment to drought stress; H_2_S and H_2_O_2_, as redox signaling molecules, are also involved in stomatal movement and stress. However, there is relatively little information on signaling and interactions among the three of them. Next, we focus our review on the interaction of ABA signaling with H_2_S and/or ROS in oxidative stress.

Under drought stress, the H_2_S-MPK4 cascade participates in both ABA- and H_2_O_2_-mediated stomatal movements, and H_2_S acts downstream of H_2_O_2_ and upstream of MPK4 in stomatal movement, although the exact location is not known [75]. However, the mechanism of how H_2_S regulates ABA signaling has been pinpointed in recent findings. ABA induces stomatal closure by means of DES1-catalyzed H_2_S production in guard cells, and H_2_S positively regulates ABA signaling via persulfidation of Open Stomata1 (OST1)/Snf1-related Protein Kinase 2.6 (SnRK2.6). Cys131 and 137 were identified to be persulfidated, which promotes the activity of SnRK2.6 and also promotes interaction with the downstream transcription factor, ABA response element-binding factor 2 (ABF2). Additionally, plants with SnRK2.6 persulfidation exhibited reduced drought resistance [57]. Interestingly, a subsequent study revealed that the persulfidation of SnRK2.6 alters its protein structure and increases the transfer efficiency of the phosphoryl group to S175, thereby increasing the level of phosphorylation [24]. Most importantly, phosphorylation and persulfidation interact in vivo. The results indicated that persulfidation of SnRK2.6 further increases the binding capacity of SnRK2.6 to ABF2, which results from the phosphorylation of S267 and S175. Meanwhile, phosphorylation of S267 may, in turn, regulate SnRK2.6 persulfidation in vivo, but it does not seem to change phosphorylation levels, which may simply be a result of structural changes in SnRK2.6 leading to potential PTM [24]. Collectively, these results indicate that persulfidation and phosphorylation form an interacting regulatory loop within the cell that modulates downstream signaling in response to ABA signaling. Thus, SnRK2.6 is definitely the intersection where ABA and H_2_S signals meet [24].

Another important target of persulfidation, Abscisic Acid Insensitive 4 (ABI4), was identified in the involvement of H_2_S and protein persulfidation in the response to ABA signaling [58]. Several lines of evidence demonstrate that ABI4 acts downstream of DES1 in *A. thaliana* to regulate the ABA response. Continued accumulation of H_2_S leads to persulfidation of the Cys250 residue of ABI4, which contributes to transcriptional activation of the MAPK cascade, thereby promoting the binding of ABI4 to the Mitogen-activated Protein Kinase Kinase Kinase 18 (MAPKKK18) promoter [58]. Surprisingly, ABI4 also binds to the DES1 promoter to activate DES1 transcription, thereby ensuring the source of cytosolic H_2_S, forming a DES1–ABI4 regulatory loop that fine-tunes ABA–MAPK signaling and stress responses [58]. Altogether, these key pieces of evidence provide the missing link in explaining how plants coordinate ABA and H_2_S signaling in response to drought, as well as providing a framework for the different PTM regulatory mechanisms induced by structural changes in proteins. Notably, how DES1 is activated in the ABA–DES1–H_2_S pathway and whether persulfidation as a redox modification regulates redox homeostasis and ABA signaling in response to drought are new questions that need to be addressed. Therefore, Shen et al. analyzed the mechanism during the ABA response in *A. thaliana* of how protein persulfidation acts in specific and reversible signaling [59]. The results indicated that ABA stimulates the persulfidation of DES1 on Cys44 and Cys205 in a redox-dependent manner, resulting in the transient accumulation of H_2_S in response to the ABA response. The sustained accumulation of H_2_S induces the persulfidation of RBOHD on Cys825 and Cys890 to promote a burst of ROS, which leads to closure of the stomata (Figure 3C). Persistent elevation of ROS levels leads to the oxidation of RBOHD and DES1 by means of persulfation to form perthiosulfinic acid, which leads to ABA desensitization. In the presence of thioredoxin (Trx), it can be reduced to -SH, thus forming a negative feedback loop that fine-tunes ABA signaling and guard cell redox homeostasis [59]. These results demonstrate that plants have a yin and yang mechanism to coordinate redox signaling and hormonal responses; H_2_S and ROS are ubiquitous in this mechanism, rapidly forming a regulatory network that allows plants to sense redox status and control ROS homeostasis in a changing environment.

## 5. Potential Tools for H_2_S and ROS Interactions: Peroxisomes and Dual Donor Systems

A recent study presented a new perspective on the relationship of H_2_S and H_2_O_2_. Confocal laser scanning microscopy (CLSM) results showed that peroxisomes of CFP-PTS1 *Arabidopsis* transgenic seedlings were randomly distributed in the form of spherical spots (green) in the root tip cells, and then an intense red fluorescence corresponding to H_2_S was observed on the spherical spots of CFP-PTS1 root tip cells using a WSP-5 fluorescent probe [15]. The authors offered the explanation that the H_2_S accumulated in *Arabidopsis* peroxisomes is likely to come from the cytoplasm and chloroplasts, because H_2_S is present in other subcellular compartments, and this lipophilic small molecule diffuses readily into other compartments [15]. Whether the peroxisomal H_2_S is endogenously generated or imported is still unknown, but it is certain that there must be some specific association of H_2_S and H_2_O_2_ in the peroxisome. The peroxisome is an H_2_O_2_-producing organelle with a large number of important H_2_O_2_-generating enzymes, of which photorespiratory glycolic acid oxidase (GOx) is one of the most critical generating enzymes. It also has a powerful antioxidant enzyme system to eliminate excess H_2_O_2_, with catalase (CAT) playing a critical role [76,77]. However, CAT and GOx can be inhibited by H_2_S though PTMs, such as persulfidation. Thus, from the present information, it is clear that H_2_S and H_2_O_2_ form a complex regulatory network in peroxisomes, with CAT as the core, which provides new insights about the mechanism of interaction between H_2_S and ROS (Figure 3D). This was the first report of the presence of H_2_S in plant or animal peroxisomes, and there are many unanswered questions that need to be further explored.

So far, although there are many relevant reports on the crosstalk between the two signaling molecules H_2_S and H_2_O_2_, which cause key biological responses (persulfidation), they are dependent on different biosynthetic pathways and signaling mechanisms. Ni et al. reported a novel dual-donor system by employing 1-thio-β-D-glucose and glucose oxidase (GOx) as substrates [78]. This enzyme system is capable of producing H_2_S and H_2_O_2_ concurrently in a slow, controlled manner without producing any byproducts that are not beneficial to biological systems. Moreover, the generation of H_2_O_2_ in this system did not affect the release of H_2_S, and the thioglucose–GOx system was effective in inducing protein persulfidation. Most importantly, thioglucose is a highly adjustable motif that can be readily tweaked to incorporate more release-control variables, allowing the delivery of H_2_S/H_2_O_2_ to be regulated by other physiologically relevant stimuli [78]. Future H_2_S and H_2_O_2_ research should benefit from this dual-release mechanism.

## 6. Conclusions and Perspectives

Over the past decades, it has been shown that almost all stress responses cause H_2_S and ROS accumulation. Therefore, it is inevitable that there is a connection between them. The accumulation of H_2_S leads to persulfidation modification of the cysteine residues of proteins, which has gradually become known as being necessary in response to oxidative stress in recent years. But this field of research is just beginning, and there are still many unanswered questions. ROS act as conserved secondary signals involved in H_2_S-mediated signaling pathways, especially H_2_O_2_, which acts as a central hub for redox signaling, a bridge for the interactions between H_2_S and other signaling molecules, and a driver of oxidative post-translational modifications. There is no doubt that PTMs are a popular topic that will continue to be researched in the coming years. Therefore, elucidating the interaction between H_2_S and ROS contributes to our insight into the function of various PTMs in oxidative stress. In this review we summarize (1) the role of H_2_S-mediated post-translational modification in oxidative stress; (2) the mechanism of interaction between H_2_S and ROS in the regulation of redox reactions and elucidate the positive cooperative role of H_2_S and ROS; and (3) the positive roles of H_2_S and H_2_O_2_ signaling in ABA-induced stomatal movement.

Furthermore, based on the available clues and evidence, we propose the following urgent questions for the future:

(1) The detection of H_2_S signaling in *Arabidopsis* peroxisomes and the fact that H_2_S regulates H_2_O_2_ metabolism by inhibiting catalase activity [15] provide a new site for H_2_S–ROS interactions. Of course, new questions follow, and whether H_2_S is produced endogenously or exogenously in peroxisomes is the next question to be urgently solved. In addition, catalase was identified in *Arabidopsis* as a target of persulfidation. However, no information exists on the specific effects of persulfidation on catalase activity. Although more and more antioxidant enzymes have been reported to be persulfidated [14], detailed insights regarding the redox-related mechanisms controlling their activity are still limited.

(2) Persulfidation-induced changes in the protein structure of SnRK2.6 established a new mechanism for intramolecular interaction between phosphorylation and persulfidation [24]. However, these findings also raise other questions as to whether persulfidation-induced protein structural changes could trigger the occurrence of other PTMs, such as S-nitrosylation, since many proteins were found to have both persulfidation and S-nitrosylation sites, requiring further biochemical and genetic validation.

(3) The proposal of a novel system with 1-thio-β-d-glucose and GOx as substrates provided new chemical tools for investigating the crosstalk mechanism of H_2_S and H_2_O_2_. Moreover, this thioglucose–GOx system was employed to effectively induce protein S-persulfidation [78]. Therefore, there are potential mechanisms of interaction between persulfuration and other Oxi-PTMs. Meanwhile, the use of this dual-release system will certainly enrich the regulatory network of H_2_S/H_2_O_2_ signaling and coactivation.

## Figures and Tables

**Figure 1 ijms-25-01935-f001:**
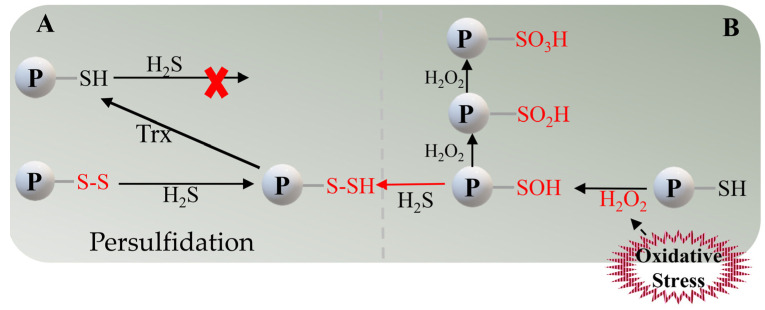
Involvement of H_2_O_2_ in H_2_S-induced persulfidation under oxidative stress. (**A**) H_2_S is unable to directly reduce cysteine residues (-SH); it accomplishes persulfation by reacting directly with the disulfide bonds (-S-S) on cysteine. (**B**) Oxidative stress induces the production of H_2_O_2_, which, in turn, oxidizes -SH to -SOH, and H_2_S can directly react with -SOH to undergo persulfation. However, a sustained increase in H_2_O_2_ concentration will continue to oxidize -SOH to -SO_2_H (sulfenylation) and will even cause further irreversible oxidization to -SO_3_H (sulfonation).

**Figure 2 ijms-25-01935-f002:**
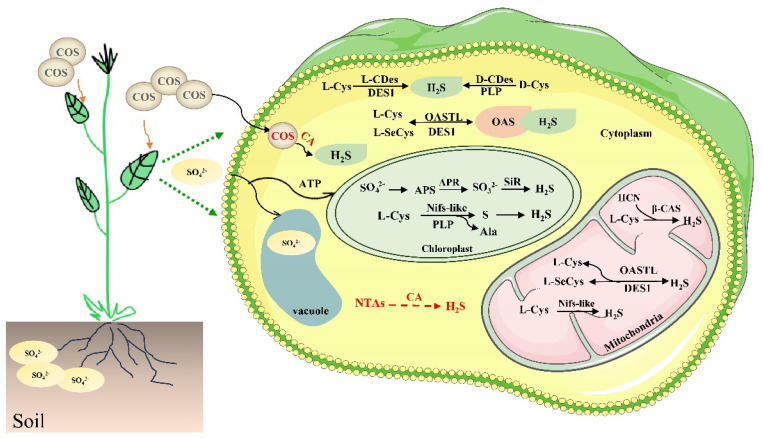
Pathways of H_2_S synthesis in plants. Endogenous synthesis of H_2_S in plant cells is mainly dependent on the root uptake of sulfate, cysteine, and carbonyl sulfide (COS). Abbreviations: APS, 5′-adenyl sulfate; APR, APS REDUCTASE; SiR, sulfite reductase; L-CDes, L-cysteine dehydrogenase; D-CDes, D-cysteine dehydrogenase; DES1, a member of the O-acetylserine sulfhydryl cleavage enzyme (OASTL) family; PLP, 5′-pyridoxal phosphate; β-CAS, β-cyanoalanine synthetase; CA, carbonic anhydrase; NTAs, acid-derived N-thiocarboxylic anhydrides. Dotted arrows indicate possible new pathways for the production of H_2_S.

**Figure 3 ijms-25-01935-f003:**
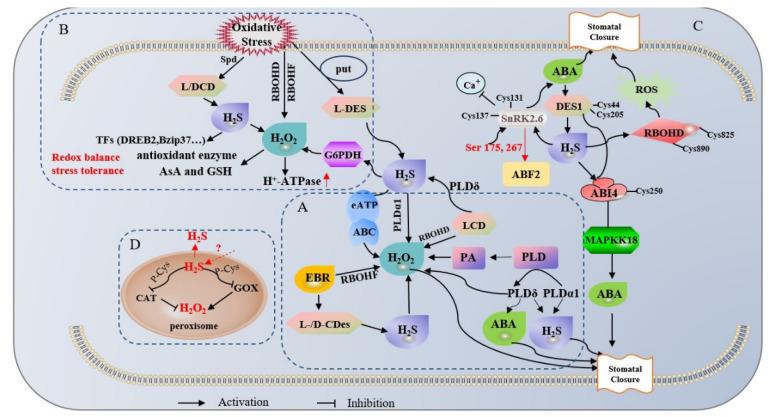
Signaling pathways and crosstalk mechanisms of H_2_S and ROS in oxidative stress. (**A**) H_2_S and H_2_O_2_ play positive roles in stomatal closure and the maintenance of redox homeostasis by regulating multiple signaling pathways. (**B**) Oxidative stress leads to the accumulation of intracellular H_2_S and ROS, triggering the production of H_2_S by means of desulfurizing enzyme activities, such as that of L-C/Des. (**C**) H_2_S and/or H_2_O_2_ positively regulate stomatal closure through post-translational modification mechanisms, such as protein persulfuration in the ABA signaling pathway, forming a complex regulatory loop with the ABA signaling pathway. (**D**) Potential sites of H_2_S and H_2_O_2_ interactions. H_2_S signaling is detected in peroxisomes, whereas CAT and GOx can be inhibited by H_2_S through PTM actions, such as persulfuration, thus maintaining redox homeostasis. Red upward arrows represent release pathways, red dashed arrows represent uncertain pathways.

## Data Availability

All data are displayed in the manuscript.

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
