# Peer review of "Hydrogen Sulfide in the Oxidative Stress Response of Plants: Crosstalk with Reactive Oxygen Species"

_ijms, 2024, doi:10.3390/ijms25031935_

Round 1
Reviewer 1 Report
Comments and Suggestions for Authors
In this review paper the authors studied the role of sulfane (H2S) in the oxidative stress response of plants. It has been found that the mechanism of action of H2S is based on the post-translational modification of the cysteine thiol group. This reaction is known as persulfidation. Since H2S cannot react with the thiol group directly, an oxidation of cysteine species by reactive oxygen species (ROS), e.g., H2O2, must first be induced. Therefore, an interaction (crosstalk) between ROS and H2S is studied. The paper is interesting; however, the novelty is not sufficiently highlighted. The paper is very textual and lacks graphical elements. The manuscript requires a revision before being reconsidered for possible publication in Int. J. Molecular Sciences. The following comments should be considered:
1.The authors should indicate the interaction between H2S, cysteine and H2O2 via a simple schematic (lines 56-58). The schematic is essential for a general understanding of the manuscript.
2.It is also recommended to include a schematic showing the H2S production pathways in plants.
3.The paper is closely related to previously published reviews on the role of H2S in regulation of oxidative stress in plants, e.g., https://doi.org/10.1093/jxb/eraa093 and https://doi.org/10.1111/jipb.13022. Therefore, the authors should sufficiently highlight the novelty of the present manuscript in comparison with previously published articles.
4.The paper is full of abbreviations. Some of them are not explained at their first mention. I recommend including an appendix at the end of the manuscript with clear definition of each abbreviation used.
5.Conclusions should be given point by point and numbered consecutively.
Comments on the Quality of English Language
The introductory sentences are not grammatically correct (lines 28-31). They should be re-arranged in the following way: “Hydrogen sulfide (H2S) is a toxic and harmful gas. The synthesis of endogenous H2S was first reported in tobacco by Harrington and Smith [1]. Since then, many studies have studied the function and properties of H2S in plants.” The introduction should be proofread by native speaker before resubmission. Furthermore, the paper is loaded with super-long paragraphs; see, e.g., lines 233-277 and 318-373. I recommend splitting these long paragraphs into shorter fractions of text to facilitate the comprehension.
Author Response
Dear reviewer,
Thank you very kindly to the editor and reviewers for reviewing and commenting on this manuscript, your comments are very important to understand this article. According to your comments, we have responded and revised point by point and highlighted them in the manuscript.
The details please see the attachment

Reviewer 2 Report
Comments and Suggestions for Authors
Manuscript ID: ijms-2818609
Type of manuscript: Review
Title: Hydrogen sulfide in the oxidative stress response of plants: Crosstalk
with reactive oxygen species
Authors: Zhiya Liu, Yayu Liu, Weibiao Liao *
This review summarized the recent development about H2S species in plants. It is considered to be well-written. However, I found some questionable points, therefore I strongly recommend strongly “Major Revisions”.
Introduction
“oxidized cysteine residues” should be noted what these specifically mean.
Overall
Some abbreviations were not normal uses, such as “superoxide dismutase” should be “SOD”. Please check them again.
Two figures were very too few. Without figures it is difficult to understand what is written.
You should insert the figures to understand what you want to say.
Comments on the Quality of English LanguageAuthor Response
Dear reviewer,
Thanks a lot for having reviewed our manuscript (ijms-2818609). We are pleased to have been given the opportunity to revise our manuscript. According to your comments and suggestions, we have carefully checked the manuscript and made the corresponding changes. The revisions have been highlighted in the revised manuscript.
The details please see the attachment

Round 2
Reviewer 1 Report
Comments and Suggestions for Authors
Authors answered my comments and improved their manuscript. It can be accepted for publication.
Reviewer 2 Report
Comments and Suggestions for Authors
Okay, these inserted figures are very helpful for the readers.
I recommended this manuscript for the publications.